# Collaboration Model for Service Clustering in Last-Mile Delivery

**Seung Yoon Ko [1], Ratna Permata Sari [2], Muzaffar Makhmudov [2] and Chang Seong Ko [2,\*]**

1   Department of Industrial Engineering, Seoul National University, Gwanak-ro, Gwanak -gu, Seoul 08826, Korea; syko1024@snu.ac.kr
2   Department of Industrial and Management Engineering, Kyungsung University, 309 Suyeong-ro, Nam-gu, Busan 48434, Korea; ratnaramalang@gmail.com (R.P.S.); muzaffar1@ks.ac.kr (M.M.)
\*   Correspondence: csko@ks.ac.kr; Tel.: +82-10-2553-4724

**Abstract:** As e-commerce is rapidly expanding, efficient and competitive product delivery system to the final customer is highly required. Recently, the emergence of a smart platform is leading the transformation of distribution, performance, and quality in express delivery services, especially in the last-mile delivery. The business to consumer (B2C) through smart platforms such as Amazon in America and Coupang in Korea utilizes the differentiated delivery rates to increase the market share. In contrast, the small and medium-sized express delivery companies with low market share are trying hard to expand their market share. In order to fulfill all customer needs, collaboration is needed. This study aims to construct a collaboration model to maximize the net profit by considering the market density of each company. A Baduk board game is used to derive the last-mile delivery time function of market density. All companies in collaboration have to specialize the delivery items into certain service clustering types, which consist of regular, big sized/weighted, and cold items. The multi-objective programming model is developed based on max-sum and max-min criteria. The Shapley value and nucleolus approaches are applied to find the profit allocation. Finally, the applicability of the proposed collaboration model is shown through a numerical example.

**Keywords:** collaboration model; express delivery service; last-mile delivery; Baduk board game; service clustering; Shapley value; nucleolus

## 1. Introduction

Business to consumer (B2C) e-commerce is becoming increasingly important in many countries in emerging markets. Globally, B2C e-commerce is a fast-growing industry and this online market was worth more than €2.5 billion worldwide in 2018. Compared to the offline market, B2C e-commerce opens up new obstacles for companies that have to manage additional issues. Some of the obstacles are high complexity of logistics operations, the small quantity with various items (i.e., 'mass customization' issues), the intangibility of electronic transactions, and quick delivery service, to name a few. In fact, many researchers believe that the most important logistical process is the last-mile distribution of order fulfillment, designed to deliver the goods ordered through the Internet to end customers [1]. Today's customers begin to demand responsiveness as an integral part of service. As consumers increasingly turn to e-commerce for all their shopping needs, a quick response service becomes a critical mission for logistic companies and retail partners across the world. The giant delivery companies as Amazon in America, Alibaba in China, and Coupang in Korea focus more on speed, safety, and accuracy. However, a fierce competition between small and medium-sized express companies means they face serious survival challenges in the rapid delivery market. Therefore, such companies are forced to restructure their delivery or service network to overcome the cost and delivery speed problems [2].

According to a holistic approach to the recent customer relationship management (CRM), the big data-enabled CRM initiatives could require several changes in the pertinent critical success factors [3]. Regarding the online customers, they are very demanding in terms of the level of service. Therefore, important attention is paid to such performance metrics as delivery times. These are punctuality, meaning getting products during certain delivery time periods, and speed of delivery, meaning amount of time spent on waiting for delivery after the order is placed by customer [4]. From the company side, last-mile delivery appears to be a part of the process of delivery that has the lowest efficiency and the largest cost because of the difficult-to-achieve target levels of service, the little order dimension, and destinations being widely dispersed [5]. The collaboration of express delivery service companies is one of the key strategic enablers in achieving common goals that directly benefit the major stakeholders. It allows companies to realize additional revenue opportunities through sharing the limited resources [6].

This study proposes a collaboration model for service clustering in last-mile delivery for participating companies and provides new perspectives to overcome potential economic downturns and survive in competitive market environments. Such collaboration between express delivery companies creates more opportunities to reduce operating costs, increase sales, and improve customer satisfaction. This study can especially provide small and medium sized delivery service companies with the opportunity for their own specialized delivery services, and it can also maximize the productivity of the last-mile delivery service that makes contact with customers in delivery process through collaboration.

The need for collaboration is recognized by most companies, but there happen three questions. First, how will they carry out collaboration using any appropriate procedures and methods? Second, how much profit can be expected through collaboration? Third, how will they fairly redistribute when there is a disparity in profits among participating companies? In this study, the above three questions will be solved at once.

The main idea is to operate only one service center that is shared by various companies in each merging region for each service class. We derive a last-mile delivery time function (LMF) using Baduk Board (also known as Go game) to maximize the profits of each participating company. A mathematical model is formulated as multi-objective programming problem for collaboration model in order to maximize the incremental profit of each participating company. In addition, this study aims to construct a sustainable collaboration model for service network design, considering service class in last-mile delivery. The success of collaboration also depends on how profit sharing and operating costs are allocated to each participating company. Hence, cooperative game theory (CGT) approaches are also applied to these companies to maintain their long-term survival.

The rest of this study is organized as follows. Section 2 clarifies the literature review related to this subject. Section 3 introduces the service class-based collaboration model concept in last-mile delivery. Section 4 includes LMF, mathematical model for multi-objective programming problem, and some approaches to solve the problem. Afterwards, the applicability of the proposed collaboration models is demonstrated through the numerical example in Section 5. Section 6 discusses three contributing issues gained from the model proposed in this study. Lastly, the study report is finalized with conclusions including future study area in Section 7.

## 2. Literature Review

A large number of previous studies in the field of express delivery service, last-mile delivery service, and collaboration model have been conducted in recent years. Yet, the source related to market density and service clustering in the last-mile delivery is still limited.

### 2.1. Last-Mile Delivery Service

Gevaers et al. [7] evaluated the density and market penetration of the delivery region creating the important effect on the efficiency in last-mile delivery. They developed some scenarios among populated urban areas and rural areas that showed the transportation cost and time window delivery could be cheaper in urban area compared to rural areas. Duin et al. [8] clearly stated that it is possible

to improve the efficiency of the last-mile delivery in B2C markets by implementing some changes in delivery processes such as changes in location, time, route, and consumer behavior. On the other hand, Agatz et al. [9] explained two ways to organize customer demand. First, determining the capacity allocation to increase the demand clustering. With demand clustering, it allowed minimizing the travel costs. Second, assigning flexible prices, needed to smoothen the customer demand. Aized and Srai provided information about last mile modeling based on a hierarchy that helped routing and congestion planning problems to deliver the goods to the end customer in a specified geographical location [10]. Regarding delivery services, Belgin et al. [11] carried out an investigation into the simultaneous pickup and delivery service as one kind of vehicle routing problem (VRP) operation. This type of operation took into account the delivery and pickup activities at the same time by the same conveyance. All items are delivered from the original depot, and all pickup items are carried back to the depot. Ozbaygin et al. [12] studied the vehicle routing problems with roaming delivery locations to find a series of the cheapest shipping routes for a fleet of capacitated vehicles and where customer orders must be sent to the customer's car trunk during the time the car was parked at one of the locations in the customer's travel plan. In addition, Tanash et al. [13] suggested that the key feature of hub-and-spoke networks was the consolidation of flows at hub facilities. They studied the modular hub location problem with single assignments. Boyer et al. [14] examined the customer density and delivery window length without offering any collaboration models.

## 2.2. Collaboration in Delivery Service

Ferdinand et al. [15] developed a decision-making model for collaboration to express courier services using a genetic algorithm-based approach for the multi-objective problem. Furthermore, Ferdinand and Ko used Shapley value allocation as coalitional game theory (CGT) to share profit allocation in express delivery services [16]. Other researchers also applied the collaboration with a regional monopoly of service centers in the field of express delivery services [17,18] and sharing of consolidation terminals among participated companies [19]. The profit allocation mechanism was also performed by Dai and Chen [20]. The methodologies to determine optimal profit-sharing allocation within sustainable collaboration were also proposed [21,22].

Furthermore, we also listed additional studies regarding last-mile delivery and collaboration as shown in Table 1 below.

**Table 1.** Related studies of last-mile delivery and collaboration.

| Category | | Researcher |
| --- | --- | --- |
| Last-mile delivery | Pricing and collaboration model using genetic algorithm in last-mile delivery services. | Ko et al. [2] |
| | An approach for estimating the end-consumer's influence of logistics system on last-mile delivery. | Galkin et al. [23] |
| | An artificial intelligent (crowdsourcing) approach for sustainable last-mile delivery. | Giret et al. [24] |
| | Utilizing innovation for last mile accessibility and connectivity to public and mass transit. | Kanuri et al. [25] |
| | Scientific approach of last mile concepts. | Clausen et al. [26] |
| | Reducing last mile side-effects on environmental by changing e-commerce customer habits. | Manerba et al. [27] |
| Collaboration | A collaboration model to increase competitiveness of companies through a monopoly of service centers and sharing of consolidation terminals. | Chung et al. [17,18] |
| | Cooperative alliance for recycling two echelon vehicle routing optimization in logistics network. | Wang et al. [28] |
| | A survey: Collaborative vehicle routing in horizontal collaboration. | Gansterer and Hartl [29] |
| | A cooperative game theoretic (CGT) approach for a logistics alliance focusing on cost savings problem. | Yea et al. [30] |
| | Two cooperation strategies for companies to analyze cost allocation in stochastic inventory models. | Timmer et al. [31] |
| | A collaboration effects of six logistics companies in Seoul. | Do et al. [32] |
| | A decision-making model for collaboration for non-collaborative versus collaborative last-mile delivery. | Villamizar et al. [33] |

Table 2 also shows a comparative analysis by extracting only those studies closely related to this study in terms of mathematical modelling of service network design and fair distribution of shared benefits in the collaboration environment of delivery services from the above-mentioned previous studies.

**Table 2.** Comparisons of previous studies and this study in collaboration environment of delivery services.

| | Service Network | | Multi-Objective | | Profit/Cost Allocation | Solution Procedure/Remark |
|---|---|---|---|---|---|---|
| | Service Region | Terminal Sharing | Max-Sum | Max-Min | | |
| Chung et al. [22] | O | | O | O | | |
| Ferdinand et al. [20] | O | O | | O | | Genetic Algorithm |
| Chung et al. [23] | O | | O | O | Shapley value | |
| Ferdinand and Ko [21] | O | O | O | | Shapley value | Co-evolutionary Algorithm |
| Chung et al. [24] | O | O | O | O | Weighted Shapley value | |
| Yea et al. [34] | Company-wide service network: Hub-spoke system | | O (Min-sum) | | Shapley value, core center, τ-value, nucleolus | |
| Ko et al. [2] | O (Last-mile delivery) | | O | O | Shapley value | Last-mile delivery |
| **This study** | O | | O | O | Shapley value, nucleolus | Service clustering |

The contribution and difference among the aforementioned collaboration models in delivery service are summarized as follows: First, service clustering problem is added to the collaboration model; second, the relation is newly derived between last-mile delivery time function and market density solved by using Baduk (Go) board game; third, the cost of movement between other participated companies is reflected in objective function, which is additionally incurred against the profits from the collaboration model; various profit allocation approaches are applied and compared based on cooperative game theory to maintain sustainable collaboration among participating companies.

## 3. Problem Statement

In general, express delivery companies handle and deliver various types of items to the end customers shown in Figure 1. This condition makes delivery service more difficult because the shipped items are still mixed, and some items require particular pieces of equipment and facilities. Therefore, the main point considered in this study is to specialize the delivery items to improve the company's competitiveness. In this study, the items are divided into certain types, which consist of regular, weighted, and cold items. In case of small and medium-sized companies that have to compete with big companies, the survival competitiveness can be improved by shipping only specialized items such as weighted or cold items. The objective of this study is to determine which of the participating companies will be responsible for which item's delivery service in candidate regions.

The collaboration model is applied as one of the survival strategies in tremendous market competition. The collaboration involves the express service companies for sharing resources and capabilities to distribute items through efficient cooperation. The participating companies have a win–win opportunity situation because they can provide better services to customers and can expect their realization to increase the net profit by utilizing their existing facilities.

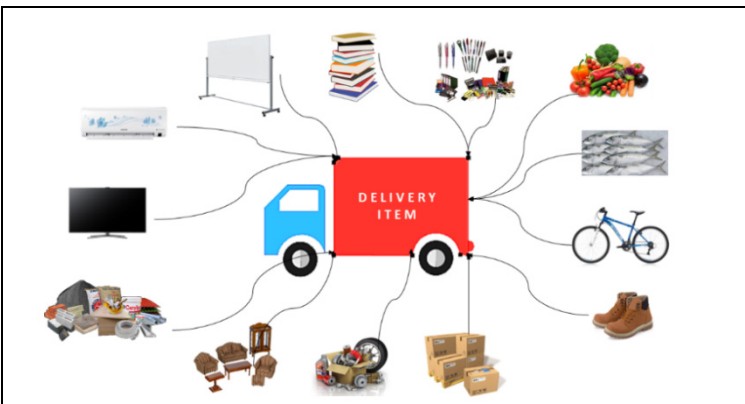

**Figure 1.** Types of items for delivery service.

To cope with substantial competition pressures, a collaboration strategy is proposed as an effective solution in most industries. By forming a collaboration system, the partners can pool their resources and strengths together in order to achieve their respective goals, share risks, gain knowledge, and gain access to new markets (Büyüközkan et al. [35]). Hitachi and Hewlett Packard went into a strategic alliance when faced with competition in the microprocessor industry to develop and to manufacture an advanced model of HP's Precision Architecture RISC MPU in order to increase the acceptance of their technology (Parise and Henderson [36]). Feng et al. [37] studied a warehouse capacity sharing problem via transshipment with a nonlinear programming model by considering fixed transshipment cost and they found that warehouse sharing with transshipment can significantly reduce the channel-wide cost.

In the case of last-mile delivery, the company under consideration is a small and medium-sized company with a low market share. Each company has a small number of delivery amounts, which have higher operational costs when compared to big delivery service companies with higher market share. Increasing profits, reducing operational costs, and expanding services are the goals emphasized in the following collaboration model.

Figure 2 shows the illustration before collaboration in region 1 to region N, and each company needs to handle three types of items, respectively. However, after the collaboration is proposed, we could see in region 1 to region N that each company only handles certain items. For instance, company A manages regular and cold items, while company B only handles the big sized/weighted items in region 1. Afterward, the company A provides delivery for cold and big sized/weighted items and company B only serves regular items in region N (see Figure 3).

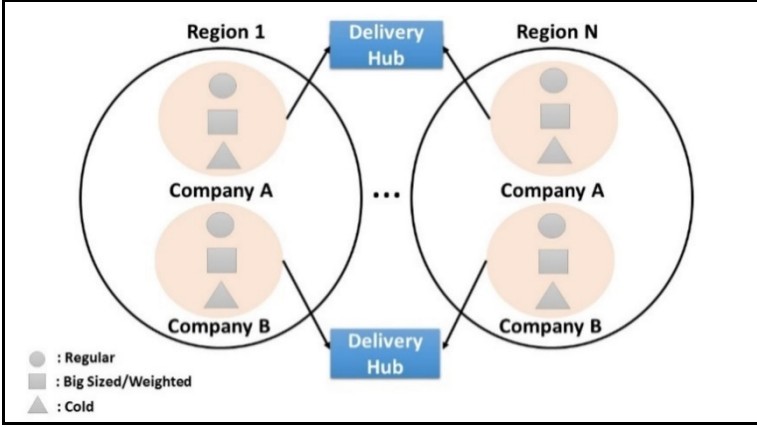

**Figure 2.** Before service class-based collaboration model.

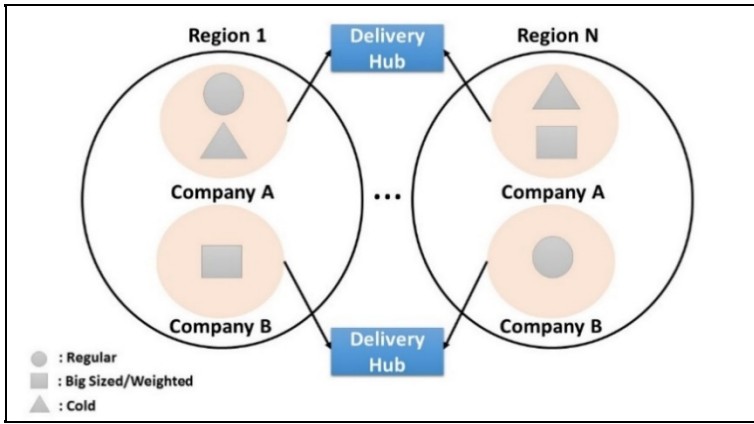

**Figure 3.** After service class-based collaboration model.

The goal of the collaboration model in last-mile delivery is to reduce costs and increase profits for each company that joins the coalition. The last-mile process is regarded as a key to customer satisfaction since it becomes the contact point at which the parcel finally arrives at the buyer's door. In particular, in this study, LMF is derived using Baduk (Go) game board to calculate the change in market share through collaboration in the last-mile delivery [2]. By applying the LMF, we could estimate average travel time among customers, average number of loads, and average unit delivery cost in succession based on market density. With unit cost function, a mathematical model is formulated as a multi-objective programming problem in order to maximize the incremental profit of each participating company.

Another important issue in collaboration problem is to maintain a long-term and sustainable alliance. To achieve this, fare profit sharing among participating companies is most important. Therefore, in this study, Shapley value and nucleolus-based allocations are applied for fair profit allocation based on cooperative game theory.

## 4. Model Design

This section provides a specific explanation related to the last-mile delivery time function, all notations, and mathematical formulation formulated as multi-objective programming. There are many methods to solve the multi-objective programming problem, including max-min and max-sum criteria. We apply the cooperative game theory for cost-saving or profit-sharing allocation. There are also several solution schemes in CGT, such as core center, τ-value, min-max core, Shapley value, and nucleolus [37,38]. Shapley value and nucleolus-based allocation are decided to solve profit sharing allocation problems. These methods are applied to find the appropriate solution in this study.

### 4.1. Last-Mile Delivery Time Function (LMF)

The last-mile delivery time function is derived to determine the travel time considering the market density. Our LMF model in this study is an extension of the previous study in Ko et al. [2].The relation between last-mile delivery time function and market density solved using Baduk (Go) board game is shown in Figure 4.

The concept is simple and acceptable, which allows us to predict the current trend in last-mile delivery service by considering the market density of the companies. In this assumption, the first move time and last move time are ignored. Table 3 shows the expected delivery times according to market density from the results of Baduk board. The equation for last-mile delivery time function, shown in Figure 5, can be expressed as:

$$t(\pi) = 2.4e^{-0.012\pi} \tag{1}$$

where *t* represents delivery time and $\pi$ is the market density.

**Table 3.** Last-mile delivery time based on Baduk board game.

| Market Density | 10 | 20 | 30 | 40 | 50 |
|---|---|---|---|---|---|
| Delivery time | 2.13 | 1.89 | 1.67 | 1.49 | 1.32 |

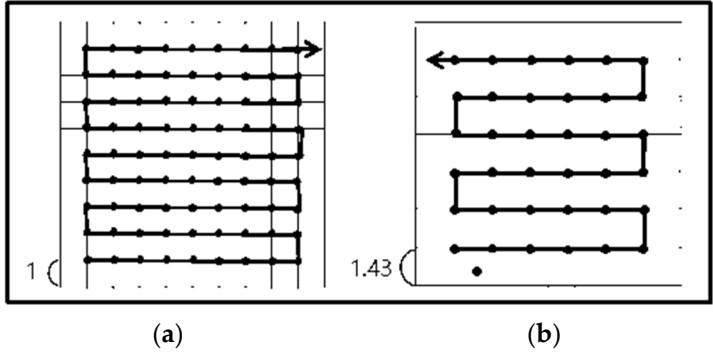

(**a**)　　　　　　　　　　　　　　　　　　　(**b**)

**Figure 4.** Baduk (Go) board for searching the last-mile delivery time function (**a**) enables us to find 81% of the market density while the delivery time is 1 min. As for (**b**), we calculate that the market density is 36% and the delivery time is 1.43 min.

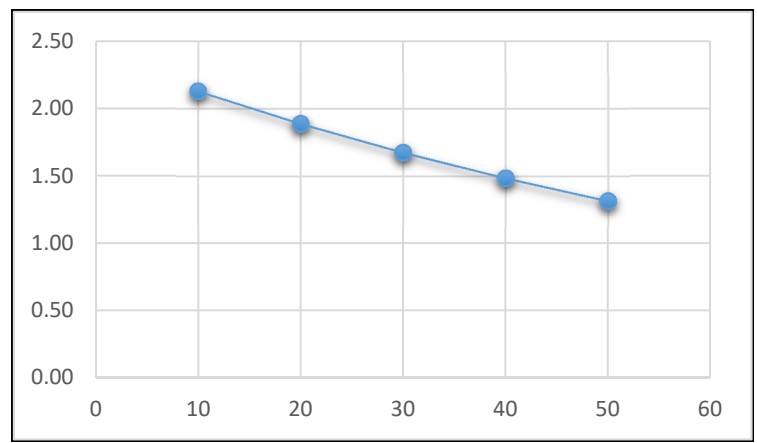

**Figure 5.** An example of last-mile delivery time function estimation based on the market density.

*4.2. Notation and Decision Variable*

This section includes all notations and decision variables that are applied in the mathematical model. The following notations are defined to formulate an optimization model for the problem:

$I$: Set of delivery service companies, $I = \{1, 2, \ldots, m\}$;

$J$: Set of merging regions, $J = \{1, 2, \ldots, n\}$;

$K$: Set of service classes, $K = \{1, 2, \ldots, k\}$;

$T_i$: Set of delivery hub for company $i, i \in I$;

$J_i^p$: Set of regions allocated to delivery hub $p$ of the company $i, p \in T_i, i \in I$;

$Q_{ip}$: Remaining capacity for processing delivery amount in delivery hub $p$ of the company $i, p \in T_i, i \in I$;

$C_{ijk}$: Unit delivery cost before alliance of the company for operating the service class $k$ of the company $i$ in region $j, i \in I, \ j \in J, k \in K$;

$C_{jk}^1$: Unit delivery cost after alliance of the company for operating the service class $k$ in region $j, j \in J, k \in K$;

$C_{jk}^2$: Unit mandated delivery cost after alliance of the company for operating the service class $k$ in region $j, j \in J, k \in K$;

$d_{ijk}$: Daily delivery amount of the company for operating the service class $k$ of the company $i$ in region $j, i \in I, \ j \in J, k \in K$;

$D_{jk}$: Total delivery amount of the company for operating the service region $j, j \in J, k \in K$;

$w_k$: Weight for handling items with service class $k$ in delivery hub;

$e_{ijk}$: Incremental cost for transit among service centers and pick-up the company for operating contributed by one unit of delivery amount with service class $k$ of the company $i$ in region $j, i \in I, j \in J, k \in K$.

Decision variable:

$x_{ijk}$: Binary variables such that $x_{ij} = 1$, if the service class $k$ of the company $i$ in region $j$ is selected, otherwise, $x_{ij} = 0, i \in I, \ j \in J, k \in K$.

### 4.3. Mathematical Model

In order to construct a mathematical model, the following assumptions are considered [2]:

1.  The market density of each express delivery company is the same in all service areas regardless of regional characteristics, which affect the delivery time of the last-mile delivery process.
2.  The level of difficulty for handling the service class is different and known.
3.  Most of the last-mile delivery services are applied in one shift/day. However, delivery services in some companies are carried out in two or three shifts a day.
4.  The daily working time for each express delivery company is the same.

In the case of the service class-based collaboration model in last-mile delivery, the concept is extended from the previous study in Ko et al. [2]. The apparent difference from the previous study is applying service class $K$ and considering the incremental cost $e_{ijk}$ incurred when moving among demand points. The proposed collaboration model is a multi-objective problem where the objective is to maximize the net profit of each participating company by forming a sustainable collaboration. In this study, we realize $C_{jk}^1 \leq C_{jk}^2 \leq C_{ijk}$, when $C_{ijk}$ means travel cost incurred by the company before collaboration and travel cost is reduced because the market density increases through collaboration. The combined cost of the three companies is defined as $C_{jk}^1$, which becomes the lower bound. In order to make a profit through collaboration, the value of $C_{jk}^2$ is specified between the lower bound and the upper bound. After applying the collaboration model, the profit of company $i$ in the grand coalition can be divided as follows. First, the incremental profit of company $i$ in the service region $j$ for its own demand is $(C_{ijk} - C_{jk}^1) d_{ijk} x_{ijk}$. In addition, the incremental profit of the company for the demands of the other companies becomes $\left(C_{jk}^2 - C_{jk}^1\right)\left(D_{jk} x_{ijk} + d_{ijk}\right)$. In contrast, if the company $i$ does not have a right to do a last-mile delivery service in any service region $j$, the incremental profit for its own demand is $(C_{ijk} - C_{jk}^2)\left(1 - x_{ijk}\right)$. Furthermore, the additional costs to move among the service centers and pick-up points become $\left(e_{ijk} x_{ijk}\right)$. The problem in this research can be explained through the following mathematical model, which consists of $m$ objective functions:

$$\text{Max } Z_1(x) = \sum_{j \in J} \sum_{k \in K}\left[\left(C_{jk}^2 - C_{jk}^1\right) w_k (D_{jk} + d_{1jk})x_{1jk} \quad + \quad \left(C_{1jk} + C_{1jk}^1 - 2C_{1jk}^2\right)w_k d_{1jk} - e_{1jk}x_{1jk} \ \right] \tag{2}$$

$$\vdots$$

$$\text{Max } Z_m(x) = \sum_{j \in J} \sum_{k \in K}\left[\left(C_{jk}^2 - C_{jk}^1\right) w_k (D_{jk} + d_{mjk})x_{mjk} \quad + \quad \left(C_{mjk} + C_{mjk}^1 - 2C_{mjk}^2\right)w_k d_{mjk} - e_{mjk}x_{mjk} \ \right]$$

$$s.t.$$

$$\sum_{i \in I} x_{ijk} = 1 \ j \in J, \ k \in K \tag{3}$$

$$L_i \ \leq \sum_{j \in J} \sum_{k \in K} x_{ijk} \ \leq \ U_i \ i \in I \tag{4}$$

$$\sum_{j \in J_i^p} \sum_{k \in K} w_k^2 \left( D_{ijk} x_{ijk} - d_{ijk} \right) \leq Q_{ip} \ p \in T_i, \ i \in I \tag{5}$$

$$x_{ijk} \in \{0, 1\} \ i \in I, \ j \in J, \ k \in K \tag{6}$$

The objective function (2) represents the net profit increase of each company. Constraint (3) provides only one service center in which each company is opened. Constraint (4) shows lower and upper bound that means the number of regions should belong to the controlled range. Constraint (5) includes the information on weight multiplication by summing the amount of pick-up and delivery amounts and by considering the processing capacity of each delivery hub. Constraint (6) includes decision variables as the binary number.

### 4.4. Max-Sum Criterion

The max-sum criterion is used to increase the total profit of each participating company within the collaboration. In order to solve the problem using the max-sum criterion, the problem can be written as follows:

$$Maximize \ Z_1 + Z_2 + \ldots + Z_m \tag{7}$$

$$s.t.$$

$$(3)–(6)$$

### 4.5. Max-Min Criterion

The max-min criterion is one method in terms of decision-making theory with the aim of maximizing minimum profits [38]. By the max-min criterion, we can expect the profit balance of each participating company. The problem in this study according to max-min criterion can be formulated as below:

$$Maximize \ \alpha \tag{8}$$

$$s.t.$$

$$(3)–(6)$$

$$Z_1 \geq \alpha \tag{9}$$

$$Z_2 \geq \alpha \tag{10}$$

$$\vdots$$

$$Z_m \geq \alpha \tag{11}$$

$$where \ \alpha = Min(Z_1, Z_2, \ldots, Z_m) \tag{12}$$

### 4.6. Shapley Value Allocation

The Shapley value allocation defined a fair way of dividing the grand coalition based on the marginal contribution of each participating company [39]. This is the concept that provides each participating company with a unique solution for profit sharing allocation. Moreover, the calculation formula stated below that expresses the cost to be allocated to each participating company (player) *i* and is based on the assumption in this study that the coalition is arranged by including the participants (players) into this coalition one at a time. Therefore, each participating company enters the grand coalition, and it is allocated to the marginal cost. This means that one participating company increases the total cost of the grand coalition. The number of participants received by this scheme depends on the order in which the participants have joined the grand coalition [22].

*4.7. Nucleolus-Based Allocation*

The nucleolus is one of the concepts to solve cost allocation problems and bankruptcy situations. This concept is based on the selection of the allocations in the least core. In the cooperative game theory (CGT), there is a core for any coalition if the three standards, which consist of completeness, rationality, and marginally, are well fulfilled. According to Frisk et al. [29], the nucleolus identifies a cost allocation that minimizes the worst inequity. For example, the individual (player) rationality should be satisfied. Hence, the aim is minimizing the dissatisfaction of each coalition.

$$Maximize\ t \tag{13}$$

$$s.t.$$

$$R_1 \geq C_1 + t \tag{14}$$

$$R_2 \geq C_2 + t \tag{15}$$

$$R_3 \geq C_3 + t\ (\text{Rationality}) \tag{16}$$

$$R_1 + R_2 \geq C_{12} + t \tag{17}$$

$$R_1 + R_3 \geq C_{13} + t \tag{18}$$

$$R_2 + R_2 \geq C_{23} + t \tag{19}$$

$$R_1 + R_2 + R_3 = C_{123}\ (\text{Completeness}) \tag{20}$$

$$R_1, R_2, R_3, t \geq 0 \tag{21}$$

## 5. Numerical Example

The last-mile delivery service is considered an important element to provide opportunities to gain a higher business to customer (B2C) market share because fast fulfillment greatly influences customer decisions. In this study, we made a decent numerical example under the assumption that there are three express delivery companies with 10 merging regions. All participating companies directly connected to the delivery hub. Market density is given, where company 1 (C1) = 5%, company 2 (C2) = 10%, and company 3 (C3) = 15% represented as the demand for each company. In last-mile delivery, the market density of each express delivery service company is the same in all service areas regardless of regional characteristics. We assumed that the time shape values for each region are generated by using the actual delivery time and the last-mile delivery time function. In particular, the actual time was used as the average value of the time collected by riding on delivery vehicles of express delivery company, which accounts for 48% of the market share in Korea.

The daily working hours are eight hours (480 min) while the revenue becomes constant, and transaction time for each delivery order is two minutes. In the grand coalition, by realizing the current market density of each participating company, we arranged the lower bound and upper bound of selected service regions as 1 and 3 for companies 1 and 3, 5 for company 2, and 5 and 7 for company 3. The data for delivery amount, last-mile delivery time, and unit delivery cost are shown in Tables 4–6. We also applied three types of items listed below in our collaboration model.

Average last-mile delivery service time is obtained by using concept of Baduk board game according to the market density. We noted that the market density increases while the travel time decreases in Table 5. This means that express delivery companies with a high market density can deliver more items to the customers.

There are two types of delivery costs, which consist of the unit delivery cost before $(C_{ijk})$ and after collaboration $(C_{jk}^1, C_{jk}^2)$. In particular, mandatory cost $C_{jk}^2$ is specified in three scenarios, which are indicated as low, middle, and highpoints. Those points aimed to calculate which points of $C_{jk}^2$ generated optimal profits for each company (see Table 6).

**Table 4.** Data for last-mile delivery amount with the market density.

| Merging Region | Last-Mile Delivery Amount ($d_{ijk}$) | | | | | | | | |
|---|---|---|---|---|---|---|---|---|---|
| | Regular Item | | | Weighted Item | | | Cold Item | | |
| | C1 (5%) | C2 (10%) | C3 (15%) | C1 (5%) | C2 (10%) | C3 (15%) | C1 (5%) | C2 (10%) | C3 (15%) |
| 1 | 83 | 167 | 250 | 42 | 84 | 126 | 28 | 56 | 83 |
| 2 | 51 | 102 | 153 | 26 | 52 | 77 | 17 | 34 | 51 |
| 3 | 87 | 174 | 261 | 44 | 87 | 131 | 29 | 58 | 87 |
| 4 | 47 | 95 | 142 | 24 | 48 | 71 | 16 | 32 | 47 |
| 5 | 82 | 164 | 245 | 41 | 82 | 123 | 27 | 55 | 82 |
| 6 | 93 | 186 | 279 | 47 | 94 | 140 | 31 | 62 | 93 |
| 7 | 60 | 120 | 180 | 30 | 61 | 91 | 20 | 40 | 60 |
| 8 | 45 | 90 | 135 | 23 | 46 | 68 | 15 | 30 | 45 |
| 9 | 56 | 113 | 169 | 28 | 57 | 85 | 19 | 38 | 56 |
| 10 | 79 | 158 | 236 | 40 | 79 | 119 | 26 | 53 | 79 |

**Table 5.** Last-mile delivery time of the market density.

| Merging Region | Time Shape | Market Density | | | | |
|---|---|---|---|---|---|---|
| | | 5% | 10% | 15% | 26.5% | 30% |
| 1 | 1.07 | 2.42 | 2.28 | 2.15 | 1.88 | 1.80 |
| 2 | 1.29 | 2.91 | 2.74 | 2.58 | 2.25 | 2.15 |
| 3 | 1.43 | 3.23 | 3.04 | 2.87 | 2.50 | 2.39 |
| 4 | 1.50 | 3.39 | 3.20 | 3.01 | 2.63 | 2.51 |
| 5 | 1.64 | 3.72 | 3.50 | 3.30 | 2.88 | 2.75 |
| 6 | 1.93 | 4.36 | 4.11 | 3.87 | 3.38 | 3.23 |
| 7 | 2.15 | 4.85 | 4.57 | 4.30 | 3.76 | 3.59 |
| 8 | 2.29 | 5.17 | 4.87 | 4.59 | 4.01 | 3.83 |
| 9 | 2.50 | 5.66 | 5.33 | 5.02 | 4.38 | 4.19 |
| 10 | 2.65 | 5.98 | 5.63 | 5.30 | 4.63 | 4.43 |

**Table 6.** Data for unit delivery cost.

| Merging Region | Unit Delivery Cost | | | Unit Delivery Cost after Collaboration | | | |
|---|---|---|---|---|---|---|---|
| | C1 | C2 | C3 | $C_{jk}^1$ (30%) | $C_{jk1}^2$ (18.75%) | $C_{jk2}^2$ (22.5%) | $C_{jk3}^2$ (26.25%) |
| 1 | 0.92 | 0.89 | 0.86 | 0.79 | 0.84 | 0.83 | 0.81 |
| 2 | 1.02 | 0.99 | 0.95 | 0.87 | 0.93 | 0.91 | 0.89 |
| 3 | 1.09 | 1.05 | 1.01 | 0.92 | 0.99 | 0.96 | 0.94 |
| 4 | 1.12 | 1.08 | 1.04 | 0.94 | 1.02 | 0.99 | 0.96 |
| 5 | 1.19 | 1.15 | 1.10 | 0.99 | 1.07 | 1.04 | 1.02 |
| 6 | 1.33 | 1.27 | 1.22 | 1.09 | 1.19 | 1.15 | 1.12 |
| 7 | 1.43 | 1.37 | 1.31 | 1.16 | 1.27 | 1.24 | 1.20 |
| 8 | 1.49 | 1.43 | 1.37 | 1.21 | 1.33 | 1.29 | 1.25 |
| 9 | 1.60 | 1.53 | 1.46 | 1.29 | 1.42 | 1.37 | 1.33 |
| 10 | 1.66 | 1.59 | 1.52 | 1.34 | 1.47 | 1.43 | 1.38 |

We performed a sensitivity analysis according to $C_{jk1}^2$, $C_{jk2}^2$, and $C_{jk3}^2$ to find the optimal solution of $C_{jk}^2$ and calculated it based on the max-sum criterion using Excel Solver. The first scenario (low point) included $C_{jk1}^2$ with 18.75% of the market density. The results of the objective function for all companies C1, C2, and C3 are $Z_1 = \$67.62$, $Z_2 = \$102.17$, and $Z_3 = \$417.57$ for which the total profit is \$587.36.

The second scenario (middle point) provided $C_{jk2}^2$ with 22.5% of the market density, then we obtained the profit for companies C1, C2, and C3. There are $Z_1 = \$137.05$, $Z_2 = \$190.91$, and $Z_3 = \$392.65$ and the total profit is \$720.61. The different results are seen in this scenario as the profit of the company

C2 earned a significant increase in its profit. The last scenario (high point) assigned $C^2_{jk3}$ with 26.25% of the market density. The companies (C1, C2, C3) gained $198.53, $290.58, and $358.54, respectively. The total profit after applying the collaboration is $847.65, which is the highest profit value compare to the previous scenarios. According to the results, we admitted that the third scenario indicated the best choice for the optimal point. Therefore, the value of $C^2_{jk3}$ with 26.25% of the market density will be used as the optimal point for further calculations. Those values are displayed through the profit variation for different value of $C^2_{jk}$ in Figure 6.

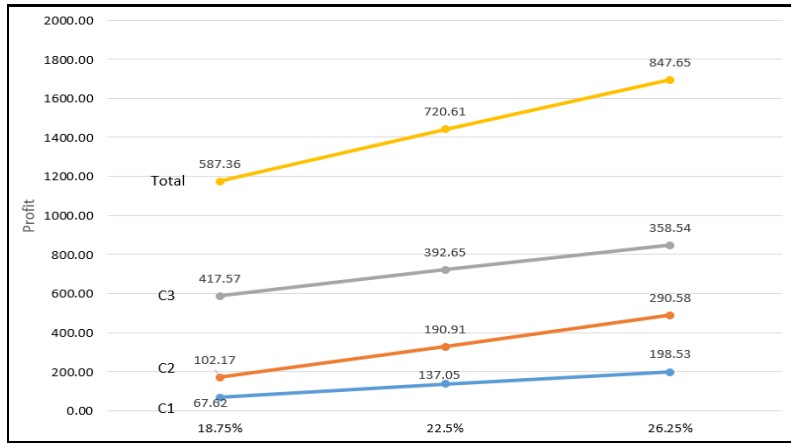

**Figure 6.** The profit variation for different value of $C^2_{jk}$.

We found the optimal solution by using the max-sum criterion as shown in Table 7. We noted that the company C1 only covered one location (region 4) for delivering regular items in the last-mile delivery service. The companies C2 and C3 are responsible almost in all regions except region 4 to do last-mile delivery service. In weighted items, the company C3 is selected in six regions. Otherwise, companies C1 and C2 are closed for last-mile delivery service activities in the same regions. Furthermore, the company C1 is only selected to be open in region 4 to serve the cold items. In contrast, C2 and C3 are closed for delivering the cold items in last-mile delivery service.

**Table 7.** Optimal solution for the max-sum criterion.

| 1. Regular Item | | | | | | | | | |
|---|---|---|---|---|---|---|---|---|---|
| Region | 1 | 2 | 3 | 4 | 5 | 6 | 7 | 8 | 9 | 10 |
| $x_{1j1}$ | 0 | 0 | 0 | 1 | 0 | 0 | 0 | 0 | 0 | 0 |
| $x_{2j1}$ | 0 | 1 | 1 | 0 | 0 | 0 | 0 | 1 | 0 | 0 |
| $x_{3j1}$ | 1 | 0 | 0 | 0 | 1 | 1 | 1 | 0 | 1 | 1 |
| **2. Weighted Item** | | | | | | | | | |
| Region | 1 | 2 | 3 | 4 | 5 | 6 | 7 | 8 | 9 | 10 |
| $x_{1j2}$ | 0 | 0 | 0 | 1 | 0 | 0 | 0 | 0 | 0 | 0 |
| $x_{2j2}$ | 0 | 1 | 0 | 0 | 0 | 0 | 0 | 1 | 1 | 0 |
| $x_{3j2}$ | 1 | 0 | 1 | 0 | 1 | 1 | 1 | 0 | 0 | 1 |
| **3. Cold Item** | | | | | | | | | |
| Region | 1 | 2 | 3 | 4 | 5 | 6 | 7 | 8 | 9 | 10 |
| $x_{1j3}$ | 0 | 0 | 0 | 1 | 0 | 0 | 0 | 0 | 0 | 0 |
| $x_{2j3}$ | 1 | 1 | 0 | 0 | 0 | 0 | 0 | 1 | 0 | 0 |
| $x_{3j3}$ | 0 | 0 | 1 | 0 | 1 | 1 | 1 | 0 | 1 | 1 |

Based on the data in Figure 7, we can summarize that overall the company C1 (horizontal line) dominated in almost every region to conduct last-mile delivery services. The company C3 is dominated

in four regions (5, 6, 7, 10) and served all types of items. Afterward, followed by the company C2 (vertical line), which dominated in two regions (2, 8), the first company (curly line) only opened in region number 4. The rest is dominated by company C2 and C3 to handle the mixed items in regions 1, 3, and 9.

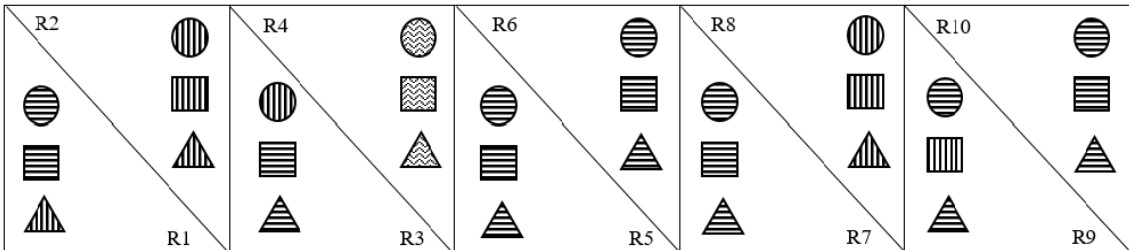

**Figure 7.** Summary result of the max-sum criterion.

The processed data using the max-min criterion are shown in Table 8. The regular items of the company C3 dominated in five regions (1, 2, 4, 7, 8). In contrast, the companies C1 and C2 are closed in the same regions. In handling for the weighted items, the company C1 opened in two regions (9, 10), the company C2 opened in three regions (5, 6, 7), and the company C3 also opened in five regions (1, 2, 3, 4, 8). In addition, those companies C1 and C2 are opened in five regions, including regions 9, 10, 2, 3, and 4 to do last-mile delivery for the cold items. At the same time, the company C3 is also selected to be open in five regions (1, 5, 6, 7, 8) for delivering cold items in last-mile delivery service. The profit for companies by using max-min criterion is as follows. The profits obtained by C1, C2, and C3 are $Z_1 = \$215.64$, $Z_2 = \$293.55$, and $Z_3 = \$296.44$. The profit result is $805.64, which is lower than the result of the max-sum criterion. Compared with the profit results obtained from max-sum criteria, the profits of companies C1 and C2 are increased. On the other hand, the profit of companies C3 is decreased from $358.54 to $296.44. In this approach, the max-min criterion indeed emphasizes the balance of profits obtained by each company that joins the collaboration.

**Table 8.** Optimal solution for the max-min criterion.

| 1. Regular Item | | | | | | | | | | |
|---|---|---|---|---|---|---|---|---|---|---|
| Region | 1 | 2 | 3 | 4 | 5 | 6 | 7 | 8 | 9 | 10 |
| $x_{1j1}$ | 0 | 0 | 0 | 0 | 0 | 0 | 0 | 0 | 1 | 1 |
| $x_{2j1}$ | 0 | 0 | 1 | 0 | 1 | 1 | 0 | 0 | 0 | 0 |
| $x_{3j1}$ | 1 | 1 | 0 | 1 | 0 | 0 | 1 | 1 | 0 | 0 |

| 2. Weighted Item | | | | | | | | | | |
|---|---|---|---|---|---|---|---|---|---|---|
| Region | 1 | 2 | 3 | 4 | 5 | 6 | 7 | 8 | 9 | 10 |
| $x_{1j2}$ | 0 | 0 | 0 | 0 | 0 | 0 | 0 | 0 | 1 | 1 |
| $x_{2j2}$ | 0 | 0 | 0 | 0 | 1 | 1 | 1 | 0 | 0 | 0 |
| $x_{3j2}$ | 1 | 1 | 1 | 1 | 0 | 0 | 0 | 1 | 0 | 0 |

| 3. Cold Item | | | | | | | | | | |
|---|---|---|---|---|---|---|---|---|---|---|
| Region | 1 | 2 | 3 | 4 | 5 | 6 | 7 | 8 | 9 | 10 |
| $x_{1j3}$ | 0 | 0 | 0 | 0 | 0 | 0 | 0 | 0 | 1 | 1 |
| $x_{2j3}$ | 0 | 1 | 1 | 1 | 0 | 0 | 0 | 0 | 0 | 0 |
| $x_{3j3}$ | 1 | 0 | 0 | 0 | 1 | 1 | 1 | 1 | 0 | 0 |

Figure 8 showed that the company C1 (curly line) dominated in two regions by using max-min criterion. Meanwhile, the companies C2 (vertical line) and C3 (horizontal line) are dominated in the rest of the regions. The company C3 (horizontal line) was only fully covered in region numbers 1 and 8.

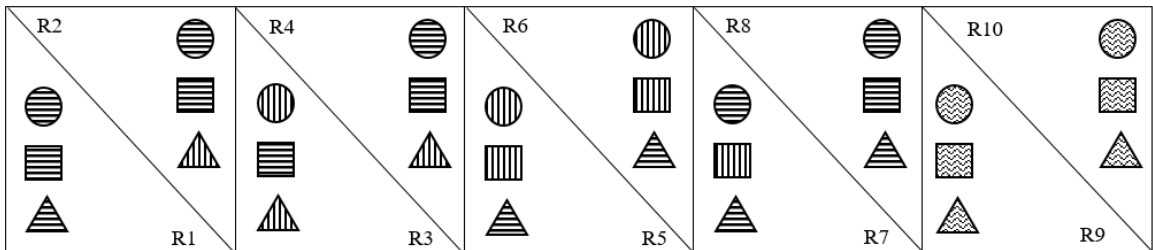

**Figure 8.** Summary result of the max-min criterion.

The result of Shapley value allocation depends on the marginal contribution of each participated company within the grand coalition. The profits sharing for $C_1$, $C_2$, and $C_3$ companies based on the Shapley value allocation are $253.02, $285.16, and $309.47, respectively. We also calculated the nucleolus-based allocation; the results obtained are different where the allocation was C1 = $354.18, C2 = $222.42, and C3 = $271.05. Compared to the allocation obtained by C2 and C3, the company C1 earned the best allocation, particularly with nucleolus-based allocation (see Table 9).

**Table 9.** Shapley value and nucleolus-based allocations.

| Combination for Collaboration | | Output | Marginal Contribution | | |
|---|---|---|---|---|---|
| | | | C1 | C2 | C3 |
| No collaboration | C1 | 0 | 0 | | |
| | C2 | 0 | | 0 | |
| | C3 | 0 | | | 0 |
| | Column Average (①) | | 0 | 0 | 0 |
| Collaboration between two companies | C1 + C2 | 576.60 | 576.60 | 576.60 | |
| | C1 + C3 | 625.23 | 625.23 | | 625.23 |
| | C2 + C3 | 689.50 | | 689.50 | 689.50 |
| | Column Average (②) | | 600.92 | 633.05 | 657.37 |
| Full collaboration | C1 + C2 + C3 (③) | 847.65 | 158.15 | 222.42 | 271.05 |
| Shapley Value | | | 253.02 | 285.16 | 309.47 |
| Nucleolus | | | 354.18 | 222.42 | 271.05 |

We compare the profits of each participating company according to max-min, max-sum, Shapley value, and nucleolus-based allocation approaches in Table 10. The Shapley value allocation specifies that the distribution of profit allocation is fair to each company based on its marginal contribution. On the other hand, applying the max-sum criterion improves the total profit in the grand coalition compared to the result of the max-min criterion. Therefore, it is better to apply the max-sum criterion from the perspective of raising the total profit. The Shapley value and nucleolus-based allocation can solve the problem on how to share coalition profits and be a good alternative in real practice. We can expect the profit balance by the max-min criterion even though with low net profit. Meanwhile, we can increase the total profit with the max-sum criterion. The profit allocation results of each participating company calculated by the Shapley value and nucleolus-based allocation can be seen in Table 10.

**Table 10.** The comparison results of max-min, max-sum, Shapley value, and nucleolus-based allocation.

|  | Max-Min | Max-Sum | Shapley Value | Nucleolus |
|---|---|---|---|---|
| C1 | $215.64 | $198.53 | $253.02 | $354.18 |
| C2 | $293.55 | $290.58 | $285.16 | $222.42 |
| C3 | $296.44 | $358.54 | $309.47 | $271.05 |
| Total profit | $805.64 | $847.65 | $847.65 | $847.65 |

## 6. Discussion

Most companies recognize the necessity of collaboration, but still have three questions for implications: What kind of procedures and methods? How much extra revenue after collaboration? How to allocate profits equitably? This study addresses practical implications that all three questions can be solved at once.

First, a strategy was used in which participating companies were divided and dedicated to the areas to be collaborated without competing. We built a model for this strategy and found the optimal partitioning policy. Table 6 shows which regions the three participating companies should monopolize according to regular, weighted, and cold items in order to maximize the profit of the coalition. On the other hand, Table 7 shows which selected regions each company should monopolize exclusively in terms of equal distribution.

Second, we developed a procedure for estimating the incremental profit generated through an increase in market share when collaboration is applied based on the current market share of each company. It led to how much the delivery cost decreased as the market share increased. For this purpose, we boarded a delivery vehicle of a real courier company to collect data and derived a last-mile delivery function using Go game board. Tables 4 and 5 show the delivery costs calculated through these procedures. Based on this, it is possible to estimate the profit generated by each company monopolizing specific target regions.

Third, the method of dividing the profits obtained through collaborations as evenly as possible according to each company's contribution was proposed for long-term sustainable collaboration. The max-sum and max-min criteria were used as the solution procedures that provide a win–win solution for each participating company. According to the result in the collaboration model, the max-min criterion generated the smallest total profit in collaboration when compared with others. The max-min criterion has limitations since it is a method of calculating the minimum profit from all collaboration. Nevertheless, this criterion could reduce the profit imbalance among participating companies. In order to find another way for increasing the total profit in this collaboration, we applied the max-sum criterion, which obtained a higher net profit than the max-min criterion. The cooperative game theory-based approaches are also utilized in this study to provide "fair" profit sharing allocation. The result value is determined through the Shapley value and nucleolus-based allocation methods.

## 7. Conclusions

Electronic commerce has been developed rapidly along with the development of information and telecommunications technology. Easy access to services via advanced technology makes e-commerce, especially business to customer, a new business model for selling the products to the market. One important key to e-commerce success through the smart platform is its reliability in last-mile delivery services. E-commerce transactions will be completed if the products ordered by customers can be delivered quickly and accurately. Delivery of goods from an express delivery company requires special handling. Understanding the characteristics of goods, delivery speed, lead time, shipping capacity, profit allocation, network system improvement, and customer satisfaction are important issues for companies providing last-mile delivery services. Meanwhile, the small and medium-sized express delivery companies encountered low market share and increasingly fierce competition at this time. The sustainable collaboration is required to survive in the rapidly changing market. In order

to improve the competitiveness of small and medium-sized express delivery companies, the service class-based collaboration model for service clustering in last-mile delivery services is proposed in this study. The multi-objective programming model is used to find efficient solutions that increased the net profit of all participating companies. An illustrative numerical example is included to verify the applicability and efficiency of the proposed collaboration model.

Through the results of this study, the following practical contributions can be expected for small and medium-sized delivery service companies: First, the foundation of a collaboration system for specialized item delivery was laid by solving service clustering; second, the incremental profit can be estimated through collaboration of companies with different market shares, which could facilitate coalition making decisions; and third, the partnership of participating companies can be continuously developed for a long time period through the cooperative game theory-based fair profit distribution.

This study also has the following academic implications: First, a mathematical model to estimate the travel time between customers using Go game was proposed. Based on this, it was possible to estimate unit delivery costs according to changes in market share. Second, service clustering was used to achieve specialization of affiliated companies. Based on this, multi-objective programming model could be developed. Third, a fair profit distribution method for the sustainability of the partnership was proposed based on the CGT. On the other hand, the limitation of this study is to consider only the travel time between vehicles for individual demands. In practice, according to the characteristics of the customer's existence, the vehicle is delivered in combination with the movement of the vehicle and on foot. In order to process these, the need to estimate the delivery time based on big data analysis is raised.

For the immediate application of this study, uncertainties in demand data should also be reflected. In a stochastic situation, robust network design should be considered in consideration of all participating companies. At this time, the market share based multi-objective programming model of this study is expected to make a significant contribution as a reference model.

In the delivery service, the demand for fresh food provided at dawn is rapidly increasing, and accordingly, many startups are entering these markets. With the working hours, the situation changes from eight hours into 24 h a day. In the past, major activities in consolidation terminals or fulfillment centers and delivery services to customers have been processed during normal working hours. Recently, 24 h processing and hour-based delivery become a new normal in delivery service areas. In this situation, a time-phased collaboration is regarded as a growth engine for small and medium-sized delivery service companies. Therefore, model development and solution provision for the time-phased collaboration can be also recommended as major research topics in the future. Additionally, the service clustering model proposed in this study will also become the reference mode in the similar problem domain and can be extended to the time dimension.

**Author Contributions:** C.S.K. applied the research ideas for last mile delivery of S.Y.K. and initiated the research project as a corresponding author. S.Y.K. developed the mathematical model and the solution procedure. R.P.S. proposed research ideas for service clustering and performed a numerical example. M.M. collected real data sets and analyzed related previous studies. All authors have read and agreed to the published version of the manuscript.

**Funding:** This research was supported by Basic Science Research Program though the National Research Foundation of Korea (NRF) funded by the Ministry of Education, Science and Technology (NRF-2018R1D1A3A03000944). This research was also supported by Kyungsung University research grants in 2019.

**Conflicts of Interest:** There is no conflict of interest.

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
