# Peer review of "Collaboration Model for Service Clustering in Last-Mile Delivery"

_sustainability, doi:10.3390/su12145844_

Round 1

Reviewer 1 Report

The authors have followed the suggested recommendations and improved the previous version

Author Response

Thank you.

Reviewer 2 Report

Manuscript ID: sustainability-736129

Title: Collaboration Model for Service Clustering in Last Mile Delivery

Review

I would like to thank you the author again for their effort in improving their work. The current document presents better addressing. However, there are still things that need to be better developed for publication.:

  • Literature review: Thank you to consider my recommendations.
  • Problem statement: Thank you to include some references that give a better understanding of the problem that you are trying to solve with this study.
  • Practical implications: I appreciate the content included however the practical implication could be more detailed. You included questions that you just answered with: “This study exposes practical implications that all three questions can be solved at once.” Please, give a deeper interpretation of your study and expose this “one solution” for 3 problems/questions. This section is the moment that you should expose and defend clearer the best and the unique value of your research to readers. I recommend you be as clear as possible and try to reach the research fundamental objective that helps small and medium-sized companies to provide a competitive delivery service.
  • Theoretical or academic implications: The content included is appreciated, however, it is not enough. This content does not explain how your paper is going to contribute from the academic point of view. How this research contributes to the scientific community or for future research on this field?

Finally, again, I appreciate the effort of the authors to improve the article, meanwhile, I suggest that the authors carefully review the paper again and improve it so that they can highlight the unique characteristics and most relevant contributions of their study.

Author Response

Dear Reviewer,

Best.

Round 2

Reviewer 2 Report

The authors have made a considerable effort to address all the reviewers’ concerns. All the changes made have helped to improve the paper. The authors have also properly refined the manuscript to refine the overall flow of the manuscript. Also, they provided a more in-depth discussion on the analysis results. Now, I consider that this paper is acceptable to be published. Finally, I recommend acceptance in its current form for publication in Sustainability.

This manuscript is a resubmission of an earlier submission. The following is a list of the peer review reports and author responses from that submission.

Round 1

Reviewer 1 Report

Dear authors,

please, find my comments as an attachment

Best regards

Author Response

Dear Reviewer,

Thank you very much for the detailed suggestions and review comments. On behalf of all the co-authors of this paper I would like to express our appreciation for the reviewers’ kind remarks, specific comments and suggestions to improve this paper, to be published in Sustainability.

Best Regards.

Reviewer 2 Report

The paper should be improved in the following aspects:

The paper should add propositions so that readers can better understand the objectives of the research.

the authors should add in the introduction what aspects this work provides with respect to other studies already carried out

A better explanation of where the data are obtained would be advisable.

Author Response

(The authors gave the same response as above.)

Reviewer 3 Report

This research proposed to construct a sustainable collaboration model for service network design, considering the express attributes in last-mile delivery on e-commerce of small and medium-sized companies.

Although the topic is interesting, I believe the document needs to be improved to be published for the following reasons:

  • The introduction is not well developed. The introduction gives repetitive information, authors should justify properly the proposed study with recent and relevant literature.
  • I appreciate all the information that the author includes in the literature review section. However, what aspects of the studies mentioned were considered and how? From the scientific point of view, authors should expand more this section and create a relation between the information included and their study.
  • Although the author uses a recent bibliography, the literature review is limited and can be better addressed.
  • On the problem statement section, some references are missing. I understand that the problem presented is a reality in different markets, however, that should be defended and the methods that will be used to make this research to solve the problem should be described properly.
  • On the conclusions, it is recommended that the authors make a better and more complete exposition of the practical implications as a fundamental objective of the present research is provide a competitive delivery service for small and medium-sized companies.
  • Additionally, I recommend that the authors include some academic implications and limitations of the research

In summary, I am grateful for had the opportunity to read this research and for the efforts of the author in carrying out research to propose a collaboration model to express delivery on a so needed market as the small and medium-sized companies. However, the current version of the study is not adequate to be published, it has been observed that the author leaves essential elements that are missing, such as a more diversified theoretical background, adequate analysis of the literature review, and include relevant contributions for academy and industry.

Therefore, I suggest that the authors carefully review the paper and improve it so that they can highlight the unique characteristics and most relevant contributions of their study.

Author Response

(The authors gave the same response as above.)

Round 2

Reviewer 1 Report

Dear reviewers,

I praise your efforts in reviewing the manuscript. You coped with my observations accurately.

Best regards

Reviewer 2 Report

The authors have improved the paper, but they do not manage to fulfill exactly the questions asked, for example, the propositions have not been added

Reviewer 3 Report

Manuscript ID: sustainability-736129

Title: Collaboration Model for Service Clustering in Last-Mile Delivery

Review

I would like to thank you the author for their effort in improving their work. However, I believe that the paper is not adequate to be published for the following reasons:

  • I appreciate all the information that the author includes in the literature review section. However, it is limited to describe other studies and it is not clear the importance of this literature to this study. How did this literature serve as a basis to this study? How these studies contribute to developing the present research.
  • Literature review: From the scientific point of view, authors should expand more this section and create a relation between the information included and their study.
  • Although the author uses a recent bibliography, the literature review is still limited and can be better addressed.
  • I do not know why authors eliminate the table 1, that table was an interesting element to the literature review that could be used as a reference on where focus a critical interpretation to try to answer questions like what is the scientific gap that you intend to fill with this study? Or how the literature review help to build the bases of this research?
  • On the problem statement section, references are still missing, and I repeat: I understand that the problem presented is a reality in different markets, however, that should be defended. Although the methods were better described, authors should describe it based on reliable literature that confirms that these methods are suitable for the circumstances presented.
  • On the conclusions, it is recommended that the authors make a better and more complete exposition of the practical implications as a fundamental objective of the present research is provide a competitive delivery service for small and medium-sized companies.
  • Additionally, I recommend that the authors include some academic implications and limitations of the research

Finally, I appreciate the effort of the authors to improve the article, meanwhile, it has been observed that the author leaves essential elements that are missing, and the empirical discoveries are not solid enough to be published in its current state.